# Factors Associated with COVID-19 Vaccine Hesitancy in Mongolia: A Web-Based Cross-Sectional Survey

**DOI:** 10.3390/ijerph182412903

**Published:** 2021-12-07

**Authors:** Davaalkham Dambadarjaa, Gan-Erdene Altankhuyag, Unurtesteg Chandaga, Ser-Od Khuyag, Bilegt Batkhorol, Nansalmaa Khaidav, Oyunbileg Dulamsuren, Nadmidtseren Gombodorj, Avirmed Dorjsuren, Pramil Singh, Gunchmaa Nyam, Dashpagma Otganbayar, Nyamsuren Tserennadmid

**Affiliations:** 1School of Public Health, Mongolian National University of Medical Sciences, Zorig Street, Ulaanbaatar 14210, Mongolia; gan-erdene@mnums.edu.mn (G.-E.A.); unurtsetseg@mnums.edu.mn (U.C.); serod@mnums.edu.mn (S.-O.K.); bilegt@mnums.edu.mn (B.B.); nansalmaa.kh@mnums.edu.mn (N.K.); oyunbileg.du@mnums.edu.mn (O.D.); nadmidtseren@mnums.edu.mn (N.G.); avirmed.d@mnums.edu.mn (A.D.); gunchmaan@gmail.com (G.N.); 2Transdisciplinary Tobacco Research Program, Loma Linda University Cancer Center, Loma Linda, CA 92354, USA; psingh@llu.edu; 3National Center for Communicable Diseases, Immunization Department, Ministry of Health, Ulaanbaatar 13335, Mongolia; dashpagam08@gmail.com; 4Department of Philosophy, National University of Mongolia, Baga Toiruu, Ulaanbaatar 14200, Mongolia; nyamsuren.ts@num.edu.mn

**Keywords:** coronavirus disease 2019, vaccine acceptance, vaccine hesitancy, severe acute respiratory syndrome coronavirus 2, Mongolia

## Abstract

Vaccine acceptance in the general public is essential in controlling the coronavirus disease 2019 (COVID-19) pandemic. The study aimed to assess the COVID-19 vaccine hesitancy in the adult population of Mongolia, and determine the associated factors. A total of 2875 individuals from urban and rural areas were recruited, and completed an online survey. Older age, urban residence, previous vaccination, high education, good knowledge of side effects, and a personal view of the importance of vaccines were associated with vaccine acceptability, whereas gender and religion were not. Receiving COVID-19 vaccine information from official government pages was related to a higher acceptance rate. Reliance on social media as a source of COVID-19 vaccine information was associated with high level of vaccine hesitancy. The side effects and the type of the COVID-19 vaccine were a major reason for hesitation. Countering false information regarding COVID-19 vaccines on social media, and promoting vaccine importance on general news websites is necessary. Moreover, providing clear and direct educational materials through official communication channels on the safety and efficacy of COVID-19 vaccines alongside information on COVID-19 symptoms, vaccine side effects, and location of vaccine administration centers among the younger populations, rural residents, and those with lower education is needed.

## 1. Introduction

As of 2 November 2021, more than 246 million cases of coronavirus disease 2019 (COVID-19) have been confirmed globally, including nearly 5 million deaths since the start of the pandemic. Countries with the most cases include the United States, Russian Federation, United Kingdom, Ukraine, and Turkey, and death rates were also high in South-East Asian countries. The Delta variant was predominant in most countries [1], and was associated with higher household transmission, hospitalizations, ICU admissions, and deaths compared with the Alpha, Beta, and Gamma variants, regardless of age or comorbidity [2,3]. Vaccination, in combination with other protective measures, is an important tool to fight against the pandemic. Since the COVID-19 outbreak, vaccine research and development efforts have begun worldwide. At the time of writing, there were 322 vaccine candidates in clinical and preclinical development, of which 128 were in the clinical phase [4]. Full immunization with all COVID-19 vaccines, which includes Pfizer-BioNTech-Comirnaty, Moderna-mRNA, Oxford-AstraZeneca, Gamaleya’s Sputnik V, and Sinovac’s CoronaVac, had high efficacy against all variants, particularly in reducing hospitalization and death rates [5]. Israel, the United Arab Emirates, the United States, and the United Kingdom were the most active countries vaccinating their citizens [6].

In Mongolia, the spread of COVID-19 infection was slow until November 2020. This was because the country took early and timely measures to prevent the spread, including closing borders and schools. From November 2020 to March 2021, cases gradually increased. However, confirmed cases accelerated in early April 2021, prompting the country to enter a 28-day lockdown. By 24 May 2021, 53,100 cases and 253 deaths had been confirmed, and a total of 2,606,563 people had received the first dose of the COVID-19 vaccine [7]. In February 2021, Mongolia received the first batch of the inactivated virus BBIBP-CorV vaccine from Sinopharm, and began immunization by administrating it to high-risk health workers. Other priority groups, including the elderly and those with chronic illnesses, were vaccinated following the arrival of further doses. In addition, the mRNA vaccine, Pfizer-BioNTech, and the non-replicating viral vector vaccines, Oxford-AstraZeneca and Gamaleya’s Sputnik V, became available for administration in the second half of 2021 in Mongolia [8]. Eventually, Mongolia successfully achieved its initial goal of vaccinating 60% of the population step-by-step, but hesitation towards vaccines in the general public remains [6]. Additionally, Malaysian experts recommend that 80–90% of the population may need to be vaccinated if there is a high cluster of infections [9].

Vaccine development has progressed rapidly, yet hesitancy to take the vaccine is a challenge around the world. Vaccine hesitancy appears to be fueled by distrust of authority, and a lack of reliable sources of information. The results of an online survey conducted among 26,852 individuals across six continents and 60 representative nations to determine knowledge, attitudes, and acceptance of a COVID-19 vaccine showed that over half of the respondents would refuse to take the COVID-19 vaccine once it is licensed, even though they strongly felt that it is crucial to get a vaccine to protect people from COVID-19 [10]. According to a national rapid assessment conducted in the US, 22% of the respondents were hesitant to take the COVID-19 vaccine. Demographic factors correlated with vaccine hesitancy included, but were not limited to, gender, ethnicity, education, income, and political affiliation [11]. The COVID-19 vaccine efficacy, origin, and consumer political characteristics are also potential factors associated with self-reported preference to accept or reject a COVID-19 vaccine among adults in the US [12]. Moreover, the source of information, cost of vaccination, and previous vaccination history may determine vaccine acceptability [13].

A pilot survey study of “Public perception, psychology, and behavior during the COVID-19 pandemic”, carried out by the School of Public Health at MNUMS in October 2020, stated that 59.8 percent of 985 respondents wanted to know more about COVID-19 treatment and vaccines. However, 46.1 percent of the respondents strongly agreed to get vaccinated if the vaccine is available and offered. This finding shows that the remaining 54 percent of the respondents are somewhat hesitant to get vaccinated. The unwillingness to get vaccinated is a serious public health issue, as a certain percentage of the population must be immunized to achieve herd immunity [13]. In 2019, WHO reported that vaccination hesitancy is one of the top 10 global health threats [14]. Investigating factors associated with vaccine acceptance is vital for creating effective public health measures to address vaccine hesitancy, and increase vaccine coverage in the population. This is essential to reduce the widespread of COVID-19 infection and its economic, healthcare, and educational burden.

In this article, we report findings from a survey which sought to determine the knowledge, attitudes, and practices of the general population in Mongolia prior to the receipt of COVID-19 vaccination. This article intends to aid health authorities and policy makers in developing an evidence-based model to promote vaccination.

## 2. Materials and Methods

### 2.1. Study Design and Population

We used a population-based cross-sectional research design to determine COVID-19 vaccine hesitancy among the public between 16 February 2021 and 25 March 2021. In Mongolia, the entire population over 18 years of age (*n* = 2,009,271) was considered as the target population towards the COVID-19 vaccine. Sampling errors, such as the non-response of individuals who refused to participate and failed to respond, were estimated at 10 percent (104 individuals). We used a simple random sampling technique to define the sampling units for the survey. A total of 2875 individuals participated in the survey, and the sample size was determined by using the following formula:n′ = (NZ^2^ P(1 − P))/(d^2^ (N − 1) + Z^2^ P(1 − P))
where n′ is the sample size with a finite population, N is the population size, Z is the Z statistic for a level of confidence, P is the expected proportion, and d is precision [15]. Mongolia is divided into four geographical regions: Western; Eastern; Central; and Khangai, and the capital city Ulaanbaatar. Each region is administratively divided into provinces, and the capital into districts. To obtain a demographically and regionally representative sample, we randomly selected one province from each region, and five districts from the capital city using online software that generates randomized results. The selected provinces included areas with and without clustering of COVID-19 cases. In the city of Ulaanbaatar, COVID-19 cases were evenly distributed.

### 2.2. Data Collection

We developed the survey questionnaire, and improved it with feedback from the Academic Board at the School of Public Health at the Mongolian National University of Medical Sciences, communicable disease and immunization experts from the World Health Organization Mongolia, and the Immunization Department of the National Center for Communicable Diseases. To facilitate participation, the questionnaire was transformed into an electronic form. At the time of data collection, the number of COVID-19 cases in Mongolia was rapidly increasing. In order to avoid crowding and increasing the risk of COVID-19 community transmission, we decided to use an online survey to collect data. The survey was anonymous, and respondents could choose to participate or not. Because Facebook is the most widely used social media platform in Mongolia, we posted a link to the survey in private groups (e.g., Residents of Khan-Uul district) to recruit participants [16]. To ensure that the same individual did not participate again, we recorded their phone number in the questionnaire, and made sure that the numbers were not repeated. If a telephone number was repeated, the result was processed by subtracting one answer after ensuring that the answers did not differ. We also indicated in the introduction part of the questionnaire that it should only be completed once. All study participants signed a written informed consent form.

### 2.3. Statistical Analysis

The research data and materials were analyzed using SPSS software version 25 (IBM, Armonk, NY, USA) by creating a database, editing, coding, recoding, and error checking. Categorical variables were presented as frequencies and percentages, and the chi-squared test was used for categorical variables. Multinomial logistic regression was performed to determine the predictors of COVID-19 vaccine hesitancy and refusal. Statistical significance was set at a *p*-value of <0.05.

## 3. Results

In Table 1, attitudes toward the COVID-19 vaccine did not vary by sex, but did vary depending on the participants’ age (*p* < 0.0001). In terms of place of residence, people who reside in an urbanized area (73.6%) had a higher tendency to be vaccinated than countryside residents (48.5%) (*p* < 0.0001). There is no association between religion and refusal of the COVID-19 vaccine. In fact, 60.2% of the research participants who consider themselves religious are vaccinated or willing to get vaccinated.

When participants’ attitudes were compared to the level of education attained (Table 2), there was a significant statistical difference (*p* < 0.0040). For instance, participants with a graduate (60.0%) and an undergraduate degree (58.4%) were most likely to get vaccinated or had been vaccinated. Only 23.1% of participants whose highest educational attainment was primary education were vaccinated or will be vaccinated, followed by participants with secondary education (51.1%).

People who received other vaccines previously had a greater willingness to be vaccinated for COVID-19 (*p* < 0.0001). Participation in mandatory vaccination was associated with an individual’s willingness to be vaccinated for COVID-19 (Table 3). Half of those who had previously been vaccinated voluntarily were willing to be vaccinated against COVID-19, whereas 62.4% of participants who had been vaccinated due to an epidemiological indication were already vaccinated or were willing to be vaccinated against COVID-19.

Vaccine acceptance was directly associated with willingness to be vaccinated for COVID-19. In other words, those who accepted all vaccines were more likely to get vaccinated (*p* < 0.001). Most participants who thought vaccines should be mandatory (80.3%) were immunized or willing to be vaccinated (Table 4), whereas half of those who believed vaccines should not be mandatory were vaccinated or willing to be vaccinated. The majority of participants who said they did not trust most vaccines were most hesitant (64.9%) about whether or not they should get COVID-19 vaccinated. In addition, more than half of those who usually accept all vaccines, but have some doubts about them (52.3%), were hesitant about COVID-19 vaccination, similar to those who accept vaccines, but sometimes refuse immunization.

In Table 5, willingness to be vaccinated for COVID-19 was defined according to participants’ source of information about the COVID-19 vaccine. The majority of participants received information about the COVID-19 vaccine from the official web page of the Mongolian Ministry of Health, and had a high willingness (73.0%) to be vaccinated or had already been vaccinated for COVID-19. Participants received the least information from FM radio, daily or weekly printed newsletters, and online newsletters and magazines. A large proportion of the study participants obtained information about the vaccine from social media platforms, i.e., Facebook and Twitter. Of them, 66.9% were willing to be vaccinated or had been vaccinated. There were similar results among participants who mainly received information about the vaccine from TV channels (66.4%). Individuals reading research articles and papers published in journals were most likely to be vaccinated (77.7%), followed by those receiving information from the official web page of the World Health Organization (WHO) (77.1%). The groups least likely to be vaccinated reported that their sources of information were from family (63.1%) and friends (63.9%), and generic local news websites, such as Ikon and Zarig (62.1%). Of those who had not received any information regarding the COVID-19 vaccine, most were willing to get vaccinated or had already been vaccinated (69%).

Among those who have been vaccinated or want to be vaccinated, the most effective preferred type of information source about the COVID-19 vaccine was infographics (76.1%), followed by posters (72.1%) and brochures (70.2%) (Table 6). Participants who preferred social media posts and podcasts as a source of COVID-19 vaccine information were the most likely to reject the vaccine (8.7%), compared with individuals who preferred other sources. Although a large proportion of participants indicated a preference for TV as a source of vaccine information, their willingness to be vaccinated was lower than those who preferred infographics (67.5%). The most needed type of information that impacted vaccine acceptance was related to the process of immunity development (71.9%), and time and place of vaccine administration (68.7%). However, the majority wanted to know more about vaccine safety, quality, consequences, and side effects, and were least concerned about the type of COVID-19 vaccine, and its storage and transportation. The individuals who wanted to know more about the vaccine’s side effects were most likely to decline COVID-19 vaccination.

As shown in Table 7, vaccination was considered one of the most effective methods to control COVID-19 (69.6%), and was directly related to a higher willingness to be vaccinated (*p* < 0.0001). Wearing face masks, hand washing, avoiding mass gathering, and social distancing were considered effective prevention methods. Surprisingly, participants who indicated that lockdown was an effective method were the least likely to get vaccinated. The prevention methods least chosen by participants were detecting and isolating infected individuals or contacts, and improving air circulation.

When assessing participants’ knowledge of symptoms, and its association with willingness to be vaccinated (Table 8), we found that the less the participants knew about COVID-19 vaccine side effects, the less likely (35.6%) they were to be vaccinated (*p* < 0.0001). However, most knew about the common COVID-19 symptoms, such as fever and chills, pain and swelling in the injection area, and headache. Seizures and anaphylactic shock were the least known possible side effects of the COVID-19 vaccine. The majority of participants who were hesitant (54.8%) or refused to be vaccinated (9.6%) reported not knowing the possible side effects of the COVID-19 vaccine.

Moreover, the perception that vaccines are essential (78.2%) was directly related to a higher willingness to be vaccinated (*p* < 0.0001) (Table 9). Participants had differing views on why vaccination is important. Those who believed that immunization protected against death or complications were most likely to be vaccinated or had been vaccinated (66.3%). Most participants thought the COVID-19 vaccine was somewhat important, whereas very few regarded it as unimportant.

In the multinomial logistic regression analysis (Table 10), there was no significant difference between male and female respondents in the proportion who hesitated or refused to be vaccinated. Participants aged 18–29 years (OR: 1.79, CI: 1.04–3.08) and participants who lived in rural areas (OR: 1.83, CI:1.32–2.53) were more likely to be hesitant to receive the COVID-19 vaccine than those aged 50 years and older, and those who lived in urban areas, age, gender, and education were adjusted for. No difference was found between age groups for vaccination refusers, but rural residence was significantly associated with COVID-19 vaccine refusal in both crude and adjusted models. The responses ‘do not accept other vaccines’, ‘do not trust other vaccines but accept some’, and ‘accept all vaccines but do not fully trust them’ were significantly associated with vaccine hesitancy and refusal when adjusted for age, gender, and education. Not knowing the importance of COVID-19 vaccines (aOR: 5.67, CI: 2.76–11.61), or regarding vaccines as unimportant (aOR: 4.13, CI: 1.33–12.79) or somewhat important (aOR: 4.84, CI: 3.40–6.90) were significantly associated with hesitancy towards COVID-19 vaccines compared to those who considered vaccination to be very important. Similar results were observed among participants who refused the COVID-19 vaccine. Participants who reported social media and general websites (aOR: 2.10, CI: 1.02–4.31) as sources of COVID-19 vaccine information were more reluctant to be vaccinated than those who received information from official health and government websites. In contrast, receiving vaccine information from friends and family members (OR: 45.41, CI: 4.17–494.67), TV, radio, and newspapers (OR: 5.50, CI: 1.79–16.89) were significantly associated with refusal to get vaccinated against COVID-19 in the adjusted model compared to those who received information from official sources. An association between primary and secondary education and refusal to vaccinate was observed, but not found among those who hesitated.

## 4. Discussion

Although vaccination is a safe and effective method to contain COVID-19 infection, and reduce the burden of disease, its success relies heavily on the willingness of the population to be vaccinated. According to a systematic review of 30 articles regarding vaccine acceptance rates among the general public in various countries around the world, Malaysia, Indonesia, Ecuador, and China have the highest vaccine acceptance rates, whereas Kuwait, Jordan, Italy, Russia, Poland, and France have the lowest acceptance rates. Furthermore, higher rates of COVID-19 vaccine acceptance were associated with sex and age [17]. Another research found that COVID-19 vaccine rejection was generally related to concerns about vaccine safety, general lack of trust, doubt regarding vaccine efficacy, and doubt about the source of the vaccine, which were the most common reasons to refuse the vaccine. In addition, women were found to be associated with lower vaccine acceptance in this study. Religious belief and a low level of education could be potential factors for vaccine hesitancy [18]. In our study, age, residence, educational level, personal acceptance of other vaccines, and personal views of vaccine importance were associated with vaccine acceptance, whereas gender and religious belief were not correlated with the vaccine acceptance rate. However, a study conducted in Bangladesh found that people living in rural areas, have low levels of education, or are elderly are more likely to refuse COVID-19 vaccination [19]. In Saudi Arabia, marital status and gender were significantly associated with vaccine acceptance [20]. The significant association between vaccine hesitancy and rural population could be explained by the sampling method, nevertheless, it is essential to extensively target rural areas, and provide information not only to reduce their doubt in vaccines, but also help them have adequate preventive methods against an infection. Also, respondents aged 18–29 years were more hesitant to get vaccinated than the oldest respondents in our study, which is consistent with a study conducted in the UK [21]. Based on this finding, public health campaigns should target young adult groups, such as bringing vaccination centers to universities to increase uptake, or providing incentives.

One of the factors influencing COVID-19 acceptance is confidence in vaccination safety, effectiveness, and side effects [22,23]. Similar to previous studies [24,25], our study showed that the main reasons for vaccine hesitancy among the general public were related to the quality, safety, and side effects of a vaccine. Common side effects include headache, fever, pain at the injection site, muscle pain, and fatigue [26]. Moreover, there is a limitation regarding breakthrough infections and hospitalizations reported in partially or fully vaccinated populations, especially those with partial vaccination. However, the COVID-19 vaccine reduced the likelihood of deaths and complications; thus, the public health benefit far outweighs the risk [27]. In addition, our study showed that people who receive information from friends and peers, and via online platforms have higher levels of skepticism. Such sources and misinformation can influence peoples’ attitudes towards vaccines, so it is crucial to provide accurate information about vaccine efficacy to gain the population’s trust with opposing perspectives [28]. For instance, information from traditional national media, such as TV or newspapers, increased the likelihood of getting vaccinated compared to information from social media channels [29]. Several reviews and studies explored the association between vaccine hesitancy and rumors or conspiracies on social media, which only spreads further in uncertain and threatening times such as the pandemic, and encourages mistrust in health officials [30,31,32]. Regulation of media content, and filtering out misinformation while raising awareness of vaccination by guiding to reliable sources could negate the effects, and increase the chances of reaching vaccine-hesitant population groups. Moreover, some studies have found a correlation between political views and vaccine acceptance [33,34]. For instance, individuals with an extreme ideology in both the left and right spectrum of political parties tend to be more skeptical about the COVID-19 vaccine [35]. Thus, future research on vaccine hesitancy could include the impacts of political viewpoints and social media, and explore effective strategies to deliver public health information to improve vaccine coverage.

There are several limitations to our study. First, because it is a cross-sectional study, it is difficult to establish a causal relationship. Also, due to the COVID-19 situation, we used an online survey, so respondents with biases may self-select and opt into the sample. However, participants were proportionally selected from each geographic region, and could be considered as representative of the general population. Finally, we recruited participants through a social media platform, thus, excluding those who do not have access to, or participate in, social networks.

## 5. Conclusions

In our study, the majority of respondents were willing to get vaccinated, and considered it a critical prevention method against COVID-19. A high proportion of participants had a reliable source of information, such as the official web page of the Ministry of Health of Mongolia, WHO, and related research articles. More than half of the participants were aware of the possible side effects of the COVID-19 vaccine. However, among those who refused to be vaccinated, the side effects and the type of the COVID-19 vaccine were significant concerns. This could be associated with the fear of vaccine effects among populations with preexisting conditions, chronic health problems, and pregnancy. Providing clear and easy-to-understand information regarding COVID-19 vaccine side effects, symptoms, and vaccine administration sites via infographics, and encouraging generic news websites and social media channels to promote awareness about the importance of COVID-19 vaccine could increase the confidence of the public in the safety and efficacy of COVID-19 vaccines, particularly in younger populations with low educational levels, and those living in rural areas.

## Figures and Tables

**Table 1 ijerph-18-12903-t001:** Willingness to be vaccinated for COVID-19 by demographics.

Variables	Vaccinated/Will Be Vaccinated		Hesitated		No, Will Not Be Vaccinated		*p*-Value
	No.	%	No.	%	No.	%	
Age group							0.0001
18–29	468	54.7%	288	33.7%	99	11.6%	
30–39	559	67.3%	214	25.8%	57	6.9%	
40–49	523	77.1%	121	17.8%	34	5.0%	
50+	413	80.7%	83	16.2%	16	3.1%	
Sex							0.5863
Male	361	67.7%	138	25.9%	34	6.4%	
Female	1602	68.4%	568	24.3%	172	7.3%	
Place of residence							0.0001
Urban	1668	73.6%	459	20.2%	140	6.2%	
Countryside	295	48.5%	247	40.6%	66	10.9%	
Religion							0.0080
Yes	218	60.2%	125	34.5%	19	5.2%	
No	321	50.1%	273	42.6%	47	7.3%	

**Table 2 ijerph-18-12903-t002:** Willingness to be vaccinated for COVID-19 by education level.

Education	Vaccinated/Will Be Vaccinated		Hesitated		Will Not Be Vaccinated		*p*-Value
	No.	%	No.	%	No.	%	
							0.0040
Uneducated	2	66.7%	1	33.3%	0	0.0%	
Primary	3	23.1%	6	46.2%	4	30.8%	
Lower secondary	29	56.9%	15	29.4%	7	13.7%	
Secondary	134	51.1%	109	41.6%	19	7.3%	
Vocational/College/	45	58.4%	24	31.2%	8	10.4%	
Undergraduate/Bachelor degree/	254	53.2%	199	41.7%	24	5.0%	
Graduate /Master’s degree and above/	72	60.0%	44	36.7%	4	3.3%	

**Table 3 ijerph-18-12903-t003:** Willingness to be vaccinated for COVID-19 by previous vaccination history.

Enrollment of Other Vaccinations	Vaccinated/Will Be Vaccinated		Hesitated		Will Not Be Vaccinated		*p*-Value
	No.	%	No.	%	No.	%	
							0.0001
Mandatory immunization	224	70.0%	86	26.9%	10	3.1%	
Immunization due to an epidemiological indication	151	62.4%	77	31.8%	14	5.8%	
Voluntary immunization	193	50.1%	171	44.4%	21	5.5%	
Supplemental immunization	60	61.2%	35	35.7%	3	3.1%	
I don’t know	61	32.4%	101	53.7%	26	13.8%	

**Table 4 ijerph-18-12903-t004:** Vaccine acceptance, personal decision, and willingness to be vaccinated for COVID-19.

Variables	Vaccinated/Will Be Vaccinated		Hesitated		Will Not Be Vaccinated		*p*-Value
	No.	%	No.	%	No.	%	
Personal belief							0.0001
Vaccine should not be mandatory	387	51.1%	327	43.1%	44	5.8%	
Vaccine should be mandatory	122	80.3%	26	17.1%	4	2.6%	
I do not know	30	32.3%	45	48.4%	18	19.4%	
Other vaccines							0.0001
Accept all vaccines	310	73.8%	102	24.3%	8	1.9%	
Accept all vaccines, but do not completely trust	125	41.9%	156	52.3%	17	5.7%	
Accept vaccines, but refuses sometimes	94	43.3%	103	47.5%	20	9.2%	
Do not trust most of the vaccines and refuse	7	18.9%	24	64.9%	6	16.2%	
Do not accept at all	3	9.7%	13	41.9%	15	48.4%	

**Table 5 ijerph-18-12903-t005:** Details of the respondents’ willingness to be vaccinated for COVID-19 compared to the source of COVID-19 vaccine information.

Current Source of COVID-19 Vaccine Information	Vaccinated/Will Be Vaccinated		Hesitated		Will Not Be Vaccinated		*p*-Value
	No.	%	No.	%	No.	%	
							0.0001
Official information source of the Government and Ministry of Health	1430	73.0%	422	21.5%	107	5.5%	
Official information source of WHO	442	77.1%	103	18.0%	28	4.9%	
Social media /Facebook, Twitter, WhatsApp/	942	66.9%	372	26.4%	94	6.7%	
Website (ikon.mn, zarig.mn, etc.)	448	62.1%	215	29.8%	58	8.0%	
Online newsletter and magazine	72	62.6%	37	32.2%	6	5.2%	
Daily or weekly printed newsletter	55	69.6%	18	22.8%	6	7.6%	
Research articles and papers published in professional journals	139	77.7%	33	18.4%	7	3.9%	
FM, Radio	37	71.2%	14	26.9%	1	1.9%	
TV channels	841	66.4%	331	26.1%	94	7.4%	
Colleagues	284	71.7%	87	22.0%	25	6.3%	
Family members	227	63.1%	98	27.2%	35	9.7%	
Friends	177	63.9%	78	28.2%	22	7.9%	
Have not received any information regarding COVID-19 vaccine	20	69.0%	7	24.1%	2	6.9%	

**Table 6 ijerph-18-12903-t006:** The various information sources compared to the respondents’ willingness to be vaccinated for COVID-19, and the type of information needed regarding COVID-19.

Variables	Vaccinated/Will Be Vaccinated		Hesitated		Will Not Be Vaccinated		*p*-Value
	No.	%	No.	%	No.	%	
Preferred type of information source to receive COVID-19 vaccine info							0.0001
Manuals	502	69.0%	181	24.9%	45	6.2%	
Brochure/leaflets	224	70.2%	74	23.2%	21	6.6%	
Flyer/poster	269	72.1%	88	23.6%	16	4.3%	
Short video	879	68.2%	338	26.2%	71	5.5%	
Infographics	372	76.1%	94	19.2%	23	4.7%	
Tweets	81	63.8%	35	27.6%	11	8.7%	
Facebook live streams, interviews, podcasts	646	63.3%	298	29.2%	76	7.5%	
TV channels	1124	67.5%	425	25.5%	116	7.0%	
FM, Radio	131	64.9%	57	28.2%	14	6.9%	
Type of information need regarding COVID-19 vaccine							0.0001
Vaccine quality	1109	65.2%	485	28.5%	106	6.2%	
Vaccine safety	1163	65.8%	494	28.0%	110	6.2%	
Vaccine storage and transportation	500	64.4%	227	29.2%	50	6.4%	
Vaccine consequences	1101	65.5%	467	27.8%	112	6.7%	
Vaccine side effects	1022	63.0%	473	29.2%	126	7.8%	
Immunity development process	896	71.9%	278	22.3%	73	5.9%	
Vaccine contraindications	768	66.6%	312	27.0%	74	6.4%	
Vaccine administration time and place	550	68.7%	210	26.2%	41	5.1%	
How to choose the vaccine type	565	64.4%	261	29.7%	52	5.9%	
Types of COVID-19 vaccines	498	62.6%	239	30.1%	58	7.3%	
Other	24	49.0%	19	38.8%	6	12.2%	

**Table 7 ijerph-18-12903-t007:** Respondents’ beliefs about effective methods of COVID-19 prevention, and their willingness to be vaccinated for COVID-19.

Four Most Effective Methods	Vaccinated/Will Be Vaccinated		Hesitated		Will Not Be Vaccinated		*p*-Value
	No.	%	No.	%	No.	%	
							0.0001
Vaccination	447	69.6%	182	28.3%	13	2.0%	
Wearing face mask	491	54.0%	357	39.3%	61	6.7%	
Lockdown	127	43.5%	144	49.3%	21	7.2%	
Social distancing	295	54.4%	213	39.3%	34	6.3%	
Improve air circulation and ventilation	192	52.5%	150	41.0%	24	6.6%	
Detecting and isolating people infected with COVID-19	139	50.9%	118	43.2%	16	5.9%	
Detecting and isolating contacted people	100	49.3%	94	46.3%	9	4.4%	
Hand washing, sanitizing, and disinfecting	325	55.1%	232	39.3%	33	5.6%	
Avoid mass gatherings	299	52.2%	242	42.2%	32	5.6%	
I do not know	3	20.0%	9	60.0%	3	20.0%	

**Table 8 ijerph-18-12903-t008:** Respondents’ knowledge of vaccine side effects measured by willingness to be vaccinated for COVID-19.

Knowledge of Side Effects	Vaccinated/Will Be Vaccinated		Hesitated		Will Not Be Vaccinated		*p*-Value
	No.	%	No.	%	No.	%	
							0.0001
General or local reactions after COVID-19 vaccination	262	61.8%	145	34.2%	17	4.0%	
Pain, swelling, and redness in the injected area	287	63.8%	144	32.0%	19	4.2%	
Loss of appetite, vomiting	102	63.0%	50	30.9%	10	6.2%	
Anaphylactic shock	50	58.1%	31	36.0%	5	5.8%	
Allergic reactions	143	57.2%	94	37.6%	13	5.2%	
Headache	230	62.8%	116	31.7%	20	5.5%	
Fever and chills	299	60.5%	171	34.6%	24	4.9%	
Shiver	98	56.0%	66	37.7%	11	6.3%	
Fatigue	134	60.4%	78	35.1%	10	4.5%	
Muscle aches	135	58.7%	84	36.5%	11	4.8%	
Seizure	35	61.4%	22	38.6%	0	0.0%	
I don’t know	100	35.6%	154	54.8%	27	9.6%	

**Table 9 ijerph-18-12903-t009:** Attitudes of survey responders on the importance of vaccinations measured by willingness to be vaccinated for COVID-19.

Variables	Vaccinated/Will Be Vaccinated		Hesitated		Will Not Be Vaccinated		*p*-Value
	No.	%	No.	%	No.	%	
Importance of vaccination							0.0001
Prevent from COVID-19 infection	363	60.1%	204	33.8%	37	6.1%	
Reduce spread of COVID-19 infection within the population	289	61.1%	160	33.8%	24	5.1%	
Develop immunity against COVID-19	307	60.4%	182	35.8%	19	3.7%	
Prevent from death and severity of COVID-19	165	66.3%	76	30.5%	8	3.2%	
Prevent from transmitting COVID-19	196	57.3%	128	37.4%	18	5.3%	
Vaccine importance							0.0001
Very important	344	78.2%	87	19.8%	9	2.0%	
Somewhat important	174	36.6%	268	56.4%	33	6.9%	
Unimportant	6	30.0%	9	45.0%	5	25.0%	
I don’t know	15	22.1%	34	50.0%	19	27.9%	

**Table 10 ijerph-18-12903-t010:** Determinants and multinomial logistic regression analysis demonstrating factors associated with hesitance and refusal of a COVID-19 vaccine.

Variables	Hesitated			Will Not Be Vaccinated				
	cOR (95% CI)	*p*-Value	aOR ^a^ (95% CI)	*p*-Value	cOR (95% CI)	*p*-Value	aOR ^a^ (95% CI)	*p*-Value
Age group								
18–29	3.26 (2.46–4.32)	<0.0001	1.79 (1.04–3.08)	0.035	5.46 (3.17–9.41)	<0.0001	1.90 (0.62–5.79)	0.261
30–39	2.00 (1.50–2.67)	<0.0001	1.43 (0.81–2.51)	0.216	2.63 (1.49–4.65)	0.001	0.76 (0.22–2.64)	0.661
40–49	1.21 (0.89–1.65)	0.231	1.14 (0.63–2.08)	0.666	1.68 (0.91–3.08)	0.095	0.65 (0.16–2.62)	0.542
50+	ref		ref		ref		ref	
Gender								
Male	1.08 (0.87–1.34)	0.499	0.83 (0.60–1.16)	0.281	0.88 (0.60–1.29)	0.505	0.71 (0.35–1.41)	0.321
Female	ref		ref		ref		ref	
Residence								
Rural	3.04 (2.50–3.71)	<0.0001	1.83 (1.32–2.53)	0.000	2.67 (1.94–3.66)	<0.0001	3.25 (1.77–5.96)	<0.0001
Urban	ref		ref		ref		ref	
Other vaccines								
Do not accept at all	13.17 (3.68–47.14)	<0.0001	4.17 (1.07–16.22)	0.039	193.75 (46.62–805.17)	<0.0001	24.34 (4.84–122.36)	<0.0001
Do not trust most of the vaccines and refuse	10.42 (4.36–24.90)	<0.0001	4.05 (1.58–10.40)	0.004	33.21 (9.08–121.48)	<0.0001	10.25 (2.36–44.49)	0.002
Accept vaccines, but refuses sometimes	3.33 (2.33–4.76)	<0.0001	1.49 (0.98–2.28)	0.072	8.24 (3.52–19.32)	<0.0001	4.68 (1.75–12.47)	0.002
Accept all vaccines, but do not completely trust	3.79 (2.74–5.25)	<0.0001	1.82 (1.23–2.66)	0.003	5.27 (2.22–12.52)	<0.0001	3.49 (1.31–9.31)	0.012
Accept all vaccines	ref		ref		ref		ref	
Vaccine importance								
I don’t know	8.96 (4.67–17.19)	<0.0001	5.72 (2.81–11.67)	<0.0001	48.41 (18.79–124.78)	<0.0001	13.30 (4.32–40.94)	<0.0001
Unimportant	5.93 (2.06–17.11)	0.001	4.50 (1.47–13.76)	0.008	31.85 (8.18–123.96)	<0.0001	10.08 (1.90–53.37)	0.007
Somewhat important	6.09 (4.50–8.24)	<0.0001	4.81 (3.38–6.84)	<0.0001	7.25 (3.39–15.49)	<0.0001	5.06 (2.12–12.10)	<0.0001
Very important	ref		ref		ref		ref	
Source of information								
Friends/Family/ Colleagues	1.34 (0.56–3.25)	0.510	5.27 (0.74–37.74)	0.095	5.61 (2.34–13.43)	<0.0001	35.88 (3.42–376.50)	0.003
TV/Radio/ Newspaper	1.78 (1.02–3.11)	0.043	1.39 (0.61–3.13)	0.419	5.52 (2.93–10.40)	<0.0001	5.34 (1.77–16.09)	0.003
Social media/Websites	1.76 (1.18–2.63)	0.005	2.11 (1.03–4.33)	0.043	1.94 (1.02–3.72)	0.044	1.68 (0.44–6.36)	0.446
Official source	ref		ref		ref		ref	
Education								
Primary	1.88 (0.59–5.99)	0.287	2.28 (0.59–8.79)	0.233	9.31 (2.37–36.66)	0.001	7.57 (1.27–45.17)	0.026
Secondary	1.02 (0.77–1.36)	0.889	1.09 (0.77–1.54)	0.624	1.86 (1.05–3.27)	0.032	3.08 (1.13–8.41)	0.028
College	0.72 (0.42–1.21)	0.209	0.91 (0.49–1.67)	0.749	2.07 (0.89–4.82)	0.092	1.41 (0.70–2.82)	0.732
Graduate	ref		ref		ref		ref	

cOR, crude odds ratio; aOR, adjusted odds ratio; CI, confidence interval; Ref, reference value. ^a^ Adjusted for age, gender, and education.

## Data Availability

Data are available upon reasonable request.

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
