# Peer review of "Factors Associated with COVID-19 Vaccine Hesitancy in Mongolia: A Web-Based Cross-Sectional Survey"

_ijerph, 2021, doi:10.3390/ijerph182412903_

Round 1
Reviewer 1 Report
The situation regarding COVID-19 keeps rapidly changing. Therefore, I would like the authors to include the latest information as possible.
- The study period of conducting the survey is not clear.
- What type(s) of vaccine(s) (mRNA, vector, inactivated…) will be administered to the people in the study population in Mongolia is not clear.
- Reports of not a few cases of breakthrough infection among highly vaccinated countries should be mentioned in Discussion as limitation which might affect vaccination policy in the future.
Author Response
Thank you for your review comments. We deeply appreciate it. Please see the attachment.

Reviewer 2 Report
The authors of the present study analysed the factors that could be related to the decision-making of whether to receive the COVID-19 vaccine. It is interesting to analyse these variables as it could set the course for public awareness policies by government agencies to increase the number of individuals with specific antibodies against the virus.
Regarding the article, some points should be clarified:
- In abstract section (page 1, line 28), the conclusions that appear in this section are imprecise and very general, without reflecting the strengths of the article. Only the need to provide clearer information to the population to increase their confidence in the vaccine is mentioned.
- In the study design section (page 2, line89) the authors reflect that they selected a total sample of 1040 participants, also considering that they did not receive a response from 104 individuals. What method did they follow to determine the representative sample size? On the other hand, it is mentioned that the target population was selected considering its geographical situation. The authors specify that the areas were selected randomly. Can the authors explain this a little better? It should also reflect how the authors have divided the sample in relation to whether they were vaccinated, whether they were hesitating to do so or not.
- In the data collection section (page 3, line 106) the authors describe that the questionnaire was reviewed by the Academic Board at the School of Public Health at Mongolian National University of Medical Sciences and by other organizations. Which were? In this same section, it is specified that an anonymous online survey was used that was distributed through social media addresses. How was this search carried out, what social media addresses were used? The survey distribution system is not very clear. This leads to the analysis of a major drawback presented by said study that must be reflected by the authors. The survey has only been distributed among individuals with social networks, without collecting information from those individuals who do not use these media. On the other hand, the study design does not guarantee that the answers of the same individual will not be repeated if he was recruited by different social networks. This information should be included in the article.
- In the statistical analysis section (page 3, line 112) a description of the analytical strategies used is made. The authors reflect that, depending on the type of variable, the results were expressed in absolute terms, percentages, means with their respective deviations, minimums, maximums. Analysing the results presented, only data is presented as a percentage. On the other hand, the chi-square test is identified as the technique used to determine if there was a significant difference or a relationship between the variables (page 3, line 118). This test only focuses on determining whether the variables are dependent or independent, and not on analysing their relationship. If you want to determine specifically what is the relationship between the variables, it is necessary to consider logistic regression to specifically detect which are the categories of variables in which it is necessary to act. In this specific case, it is recommended to rethink the statistical analysis to specifically detect the points susceptible to action. For example, it is established that there is a dependence of age with respect to vaccination, but this relationship is not deepened, nor are the target age groups for awareness programs clearly identified.
- In the conclusions section (page 9, line 289), again the authors comment on the need to implement clearer and more direct materials and campaigns to increase the confidence of vaccines. However, this is not a direct conclusion from the results presented. The conclusions of this study should be aimed directly at identifying in a concrete and specific way the population groups that show some type of need for information regarding vaccination against COVID-19.
- In the references section (page 10, line 310) many errors with the style have been detected. It is recommended that you review the journal regulations for this section.
Best regards
Author Response
Dear Reviewer,
Thank you for giving us the opportunity to submit a revised draft of our manuscript titled "Factors associated with COVID-19 vaccine hesitancy in Mongolia" to IJERPH. We appreciate the time and effort that you and the reviewers have dedicated to providing your valuable feedback and insightful comments on the manuscript. We have been able to incorporate changes to reflect the suggestions provided by the reviewers. Please see the attachment.
Thank you,
Davaalkham Dambadarjaa and co-authors of the manuscript

Reviewer 3 Report
This article reports the key findings from the survey that aimed to deterrmine the knowledge, attitudes and practices of the general population prior to COVID 19 vaccination in Mongolia. It is a very interesting study. Some points to consider:
Line 19: Better explain COVID 19 when first mention the term, line 17.
Table 7 is not shown correctly
When discussing Table 8 results, the 54.8% and the 9.6% of the ones hesitated and not willing to be vaccinated accordingly should also be mentioned as the % is the highest for the people not knowing the symptoms after vaccination.
Line 266 please correct punctuation
Author Response

(The authors gave the same response as above.)

Round 2
Reviewer 1 Report
(There are no comments)
Author Response
Dear Reviewer 1,
Thank you very much for your review and for your attention. Your suggestions were greatly important and made our paper better. Thank you one more time for your attention.
Sincerely,
Prof Davaalkham Dambadarjaa
Reviewer 2 Report
The authors have answered some doubts raised in the previous review. However, some points remain to be clarified.
It is specified that the survey was distributed through Facebook. It would be important to include this information in the abstract or even in the title, clearly indicating that the results are derived directly from responses through a social media.
In relation to statistical analysis, it is necessary to delve into the techniques carried out for logistic regression. It is convenient that this analysis derives directly from what is observed in the rest of the results, and that it is only carried out when statistical significance has been observed in the chi-square test. It is recommended that an in-depth analysis be carried out, multinominal logistic regression allows classifying the subjects of the sample based on the value taken by the different predictive variables. In this way, the characteristics of the target populations will be clearly identified for the elaboration of specific informative programs.
Best regards
Author Response
Dear Reviewer 2,
We are very grateful for your comments and suggestions on our manuscript. They certainly allowed us to improve our paper and help it become more impactful. Please see the attachment for related changes.
We hope that you will find our manuscript acceptable in its present form. Again, thank you very much for your attention.
Sincerely,
Prof Davaalkham Dambadarjaa
